# Dynamics of Polar-Core Spin Vortices in Inhomogeneous Spin-1 Bose-Einstein Condensates

**Zachary L. Stevens-Hough[1], Matthew J. Davis[1] and Lewis A. Williamson[1]**

**1** ARC Centre of Excellence for Engineered Quantum Systems, School of Mathematics and Physics, University of Queensland, St Lucia, Queensland 4072, Australia

⋆ lewis.williamson@uq.edu.au

## Abstract

In the easy-plane phase, a ferromagnetic spin-1 Bose-Einstein condensate is magnetized in a plane transverse to the applied Zeeman field. This phase supports polar-core spin vortices (PCVs), which consist of phase windings of transverse magnetization. Here we show that spin-changing collisions cause a PCV to accelerate down density gradients in an inhomogeneous condensate. The dynamics is well-described by a simplified model adapted from scalar systems, which predicts the dependence of the dynamics on trap tightness and quadratic Zeeman energy. In a harmonic trap, a PCV accelerates radially to the condensate boundary, in stark contrast to the azimuthal motion of vortices in a scalar condensate. In a trap that has a local potential maximum at the centre, the PCV exhibits oscillations around the trap centre, which persist for a remarkably long time. The oscillations coincide with the emission and reabsorption of axial spin waves, which reflect off the condensate boundary.

# 1   Introduction

Topological defects underpin a plethora of physical processes, including ordering dynamics [1], turbulence [2] and topological phase transitions [3]. The interplay between density inhomogeneity and topological defects provides a rich area of exploration, giving rise to striking effects such as defect pinning [4–11], modified critical dynamics [12–15] and novel phases of matter [16–18]. Spin-1 Bose-Einstein condensates support multiple spin phases and associated topological defects [19], are well isolated from the environment, and can be manipulated with high precision. This, combined with non-destructive imaging techniques [20], has enabled *in situ* experimental studies of defect interactions [21–23] and the role of defects in the early stages of phase ordering [24].

In the easy-plane phase a ferromagnetic spin-1 condensate is magnetized in the transverse plane with SO(2) spin symmetry [25]. In two dimensions circulation of transverse spin, with no additional global phase circulation, gives rise to polar-core spin vortices (PCVs) [24, 26]. These play a fundamental role in both equilibrium [27–29] and non-equilibrium properties [24, 30–37] of the system. A point-vortex model of PCVs shows that opposite (same) charged PCVs tend to attract (repel), which is attributable to spin-changing collisions in the vortex core [38–40]. To date, theoretical studies of PCV dynamics have focused on homogeneous spinor condensates. On the other hand, spinor condensates realised experimentally have inhomogeneous densities [24, 41–44]. It is well known that a vortex in a scalar condensate is sensitive to the background condensate profile induced by a trapping potential, which causes vortex procession in an axisymmetric trap [45–56]. The corresponding question in regards to PCVs — how does a single PCV move in an inhomogeneous density profile — has not been explored to our knowledge.

In this work we show how PCVs move in inhomogeneous density profiles, focusing on axisymmetric traps. We show that a PCV tends to move down density gradients, irrespective of the PCV charge. Hence in a harmonic trap a PCV moves radially outward, in stark contrast to the azimuthal motion of a scalar vortex. This radial motion is associated with a separation ("stretching") of the component circulations that make up a PCV, which tend to move azimuthally in the background field but are bound by the spin exchange energy. A trap with a local potential maximum at the centre produces a local minimum in condensate density. An off-center PCV then moves radially inward. Rather than settling at the trap centre, we find that the PCV instead shows periodic perturbations in displacement, coinciding with the emission and reabsorption of axial spin-wave excitations that reflect off the condensate boundary. These dynamics are remarkably long-lived, decaying as a slow power law in time for long times. We analyse the dynamics of incompressible and compressible energy during the PCV dynamics. We find that the incompressible kinetic energy of a PCV decreases as the vortex moves to lower densities. In the harmonic trap this energy is primarily converted into internal spin energy. In the trap with a local potential maximum at the centre the energy is converted back-and-forth between compressible and incompressible kinetic energy, consistent with the emission and reabsorption of axial spin waves.

This work is organised as follows. In Sec. 2 we introduce the system and PCVs. In Sec. 3 we present a general model of PCV dynamics in an inhomogeneous background. We then present numerical results of PCV dynamics in a harmonic trap (Sec. 4) and in a box trap with a local potential maximum at the centre (Sec. 5), comparing with the model in Sec. 3. In Sec. 6 we analyse the energy exchanges that occur during the dynamics. We conclude in Sec. 7.

## 2  Polar-core spin vortices

A spin-1 condensate can be described by a three component spinor field $\Psi = (\psi_{-1}, \psi_0, \psi_1)^T$, with $\psi_m$ describing atoms in the magnetic sublevel $m \in \{-1, 0, 1\}$. In a quasi-2D geometry in the mean-field approximation, the system is described by the Hamiltonian [57]

$$H = \int d^2\mathbf{r} \left\{ \sum_{m=-1}^{1} \psi_m(\mathbf{r})^* \left[ -\frac{\hbar^2}{2M}\nabla^2 + U(\mathbf{r}) + qm^2 \right] \psi_m(\mathbf{r}) + \frac{g_n}{2}n(\mathbf{r})^2 + \frac{g_s}{2}|\mathbf{F}(\mathbf{r})|^2 \right\}. \quad (1)$$

Here $n(\mathbf{r}) = \sum_m \psi_m^* \psi_m = \Psi^\dagger \Psi$ is the areal number density and $\mathbf{F}(\mathbf{r}) = \sum_{\mu=x,y,z} \Psi^\dagger f_\mu \Psi \hat{\mathbf{s}}_\mu$ is the areal spin density with $f_\mu$ the spin-1 Pauli matrices in the direction $\hat{\mathbf{s}}_\mu$. The quadratic Zeeman energy $q$ controls the single-particle level spacings along the quantization axis $\hat{\mathbf{s}}_z$, and can be tuned in experiments using a variety of techniques [58]. The linear Zeeman shift has been removed by transforming $\Psi$ into a frame rotating at the Larmor frequency. The spin and density interaction strengths are $g_s$ and $g_n$ respectively. The ratio of $q$ to spin interaction energy results in a rich variety of magnetic ground states [59, 60]. We consider ferromagnetic spin interactions, $g_s < 0$, as realised in $^{87}$Rb [61, 62] and $^7$Li [43] condensates.

The time-evolution of each spin level is governed by the spin-1 Gross-Pitaevskii equations [57]

$$i\hbar \frac{\partial \psi_m}{\partial t} = \left[ -\frac{\hbar^2}{2M}\nabla^2 + U(\mathbf{r}) + qm^2 + g_n n \right] \psi_m + g_s \sum_{m'=-1}^{1} \mathbf{F} \cdot f_{mm'} \psi_{m'}. \quad (2)$$

We take $g_n/|g_s| = 10$ so that the dynamics are predominantly confined to the spin degrees of freedom. The spin time $t_s \equiv \hbar/q_0$ and the spin healing length $\xi_s \equiv \hbar/\sqrt{Mq_0}$ are then convenient time and length scales. Here $q_0 = 2|g_s|n_0$ with $n_0 \equiv n(\mathbf{0})$ the condensate density at the trap centre $\mathbf{r} = \mathbf{0}$.

In a uniform system, $q = q_0$ is a quantum critical point separating the unmagnetized ("polar") phase ($q > q_0$) from the magnetized phases ($q < q_0$) [60]. When $0 < q < q_0$, the ground state of Eq. (1) is magnetized in a plane perpendicular to the external Zeeman field, termed the easy-plane phase. The ground state manifold consists of both a U(1) global gauge symmetry and an SO(2) spin symmetry $e^{-if_z\varphi}$ [25]. A $2\pi\kappa$ ($\kappa \in \mathbb{Z} \setminus \{0\}$) phase winding of $\varphi$ gives rise to circulations $2\pi m\kappa$ in the spin components $\psi_m$, and a $2\pi\kappa$ circulation of transverse spin $F_\perp = F_x + iF_y$. With no additional global phase winding, this circulation can be identified as a polar-core spin vortex, with vortex core filled by the (circulation-free) $m = 0$ component [24, 26],

$$\Psi(\mathbf{r}) = \sqrt{\frac{n}{2}} \begin{pmatrix} \sin\beta\, g_1(\mathbf{r})e^{-i\kappa\phi(\mathbf{r})} \\ \sqrt{2}\cos\beta\, g_0(\mathbf{r}) \\ \sin\beta\, g_1(\mathbf{r})e^{i\kappa\phi(\mathbf{r})} \end{pmatrix}. \quad (3)$$

Here $\phi(\mathbf{r}) = \text{phase}(z)$ is the phase of the complex number $z = x + iy$, $g_m(\mathbf{r})$ accounts for the density of the vortex core, with $g_1(\mathbf{0}) = 0$, and $\cos 2\beta = q/q_0$. A notable feature of Eq. (3) is the equal and opposite phases of the $\psi_{\pm 1}$ components. This ensures that the spin exchange component $E_{\text{se}}$ of the spin interaction energy is minimised, with

$$E_{\text{se}} = g_s \int d^2\mathbf{r} \left[ \psi_0^2 \psi_1^* \psi_{-1}^* + (\psi_0^*)^2 \psi_1 \psi_{-1} \right]. \quad (4)$$

The spin interaction energy is responsible for spin-changing collisions $0 + 0 \leftrightarrow 1 + (-1)$ and is absent in multi-component condensates of distinct species or distinct hyperfine manifolds.

In a trapped (inhomogeneous) system the ground state is easy-plane for $q \lesssim 2|g_s|n(\mathbf{r})$ and hence the phase depends on the inhomogeneous condensate density. We will consider traps

with a single phase boundary at $r = r_{F_\perp}$, such that the system is easy-plane for $r < r_{F_\perp}$ and polar for $r > r_{F_\perp}$. Approximating $n(\mathbf{r})$ by the Thomas-Fermi density [63],

$$n_{\mathrm{TF}}(\mathbf{r}) = n_0 \left( 1 - \frac{U(\mathbf{r})}{\mu} \right), \tag{5}$$

we define $r_{F_\perp}$ via $U(r_{F_\perp}) = \mu(1 - q/q_0)$, with $\mu = g_n n_0$.

## 3 A model of PCV dynamics in an inhomogeneous condensate

To gain an intuitive understanding of how a PCV moves in an inhomogeneous density, we formulate a model adapted from scalar systems [64–67]. A PCV consists of equal and oppositely charged vortices in the $m = \pm 1$ components, see Eq. (3). Following [67], we write the wavefunction for these vortex fields as

$$\psi_m(\mathbf{r}, t) = (z(\mathbf{r}) - z_m(t)) A_m(\mathbf{r}, t) e^{i\theta_m(\mathbf{r}, t)}. \tag{6}$$

The complex phase factor $z - z_m$, with $z_m = x_m + im\kappa y_m$ the centre of circulation and $z(\mathbf{r}) = x + iy$, accounts for both a circular vortex phase circulation and the density dip at the vortex core $z_m$. The background amplitude $A_m$ and phase profile $\theta_m$ describe the remaining portion of $\psi_m$. The amplitude $A_m$ includes the inhomogeneous density caused by the trap, while the background phase $\theta_m$ arises from any imposed background flow as well as enforcing the condition of zero particle flow across the condensate boundary [68]. Additionally, and unique to the spinor system, $A_m$ and $\theta_m$ may be affected by separation of the $\psi_{\pm 1}$ vortices due to spin-exchange interactions [40, 69].

An equation of motion for $\mathbf{x}_m$ can be obtained by evolving the state (6) infinitesimally using Eq. (2) and identifying the new zero point in density. Following the method in [67], this gives

$$\dot{\mathbf{x}}_m = \frac{\hbar}{M} \left[ \kappa m \hat{\mathbf{z}} \times \nabla \ln A_m - \nabla \theta_m \right]\big|_{\mathbf{x}_m} + \mathbf{J}_m, \tag{7}$$

where

$$\mathbf{J}_m = \frac{g_s}{\hbar} \left( -\mathrm{Im} \frac{\psi_0(\mathbf{x}_m)^2 \psi_{-m}^*(\mathbf{x}_m)}{A_m(\mathbf{x}_m) e^{i\theta_m(\mathbf{x}_m)}}, \mathrm{Re} \frac{\psi_0(\mathbf{x}_m)^2 \psi_{-m}^*(\mathbf{x}_m)}{A_m(\mathbf{x}_m) e^{i\theta_m(\mathbf{x}_m)}} \right). \tag{8}$$

The first term in Eq. (7) takes the same form as that of a vortex in a scalar condensate. In an axisymmetric trap with no imposed background flow, this term induces azimuthal vortex motion [67]. The second term $\mathbf{J}_m$ is a source term arising from the spin exchange component of the spin interaction energy, Eq. (4), and is the key feature that will distinguish the PCV dynamics. This source depends on the "stretch coordinate" $\mathbf{s} = \mathbf{x}_1 - \mathbf{x}_{-1}$ via

$$\psi_{-m}(\mathbf{x}_m, t)^* = \psi_m(\mathbf{x}_m - m\mathbf{s}, t). \tag{9}$$

In the absence of $\mathbf{J}_m$ and any background superfluid flow, the $m = \pm 1$ vortices will circulate the trap centre in opposite directions, causing a separation of the $m = \pm 1$ vortex cores. In the presence of $\mathbf{J}_m$, this "stretching" cannot occur indefinitely, as the components are bound together by the spin exchange energy [38, 39]. To lowest order in the stretch coordinate, Eq. (9) gives $\psi_{-m}(\mathbf{x}_m)^* \approx -m\mathbf{s} \cdot \nabla \psi_m|_{\mathbf{x}_m}$. This gives

$$\mathbf{J}_m \approx \frac{|g_s| \psi_0(\mathbf{x}_m)^2}{\hbar} \hat{\mathbf{z}} \times \mathbf{s}, \tag{10}$$

where we have taken $\psi_0$ to be real without loss of generality.

Rewriting Eq. (7) in terms of $\mathbf{s}$ and the PCV position $\mathbf{x}_v = (\mathbf{x}_1 + \mathbf{x}_{-1})/2$, we obtain the equations of motion

$$\dot{\mathbf{x}}_v \approx \frac{1}{2} \sum_m \mathbf{J}_m, \tag{11}$$

$$\dot{\mathbf{s}} = \frac{\hbar \kappa}{M} \left[ \hat{\mathbf{z}} \times \nabla \ln A - \nabla \theta \right] \Big|_{\mathbf{x}_v} + \frac{1}{2} \sum_m m \mathbf{J}_m. \tag{12}$$

Here $\nabla \theta|_{\mathbf{x}_v} \equiv (\nabla \theta_1|_{\mathbf{x}_1} - \nabla \theta_{-1}|_{\mathbf{x}_{-1}})/2$ and $\nabla \ln A|_{\mathbf{x}_v} \equiv (\nabla \ln A_1|_{\mathbf{x}_1} + \nabla \ln A_{-1}|_{\mathbf{x}_{-1}})/2$. We have also made the approximations $\nabla \theta_1|_{\mathbf{x}_1} \approx -\nabla \theta_{-1}|_{\mathbf{x}_{-1}}$ and $\nabla A_1|_{\mathbf{x}_1} \approx \nabla \ln A_{-1}|_{\mathbf{x}_{-1}}$. Utilising Eq. (10) with $\psi_0(\mathbf{x}_m) \approx \psi_0(\mathbf{x}_v)$ we find

$$\ddot{\mathbf{x}}_v \approx \frac{1}{2} \sum_m \dot{\mathbf{J}}_m \approx \frac{|g_s| \psi_0(\mathbf{x}_v)^2}{\hbar} \hat{\mathbf{z}} \times \dot{\mathbf{s}}, \tag{13}$$

$$\dot{\mathbf{s}} \approx \frac{\hbar \kappa}{M} \left[ \hat{\mathbf{z}} \times \nabla \ln A - \nabla \theta \right] \Big|_{\mathbf{x}_v}. \tag{14}$$

We have neglected the term proportional to $\dot{\psi}_0(\mathbf{x}_v)$ in Eq. (13) since it is proportional to $\mathbf{s}$ and hence small. Equation (14) shows that density and phase gradients will tend to stretch a PCV and Eq. (13) shows that this stretching will cause a PCV to accelerate in a direction orthogonal to the stretching. We can interpret the term $(1/2) \sum_m \dot{\mathbf{J}}_m$ in Eq. (13) as a "force per unit mass" acting on the vortex, which depends on the dynamics of $\mathbf{s}$.

Equation (14) has limited utility as a direct computational tool, since the calculation of $\nabla \theta$ is not straightforward [67]. In addition to this, the dynamics of the stretch coordinate is susceptible to damping [40]. In a scalar system, an analytic calculation gives

$$\nabla \theta \Big|_{\mathbf{x}_v} \approx \hat{\mathbf{z}} \times \nabla \ln A \ln |\xi_R \nabla \ln A| \Big|_{\mathbf{x}_v}, \tag{15}$$

where $\xi_R \sim \xi_s$ acts to regularise the otherwise divergent background current field [55,65,67]. Hence, crudely, $\dot{\mathbf{s}} \sim (\hbar \kappa / M) \hat{\mathbf{z}} \times \nabla \ln A$. Approximating $A$ by the Thomas-Fermi amplitude $\sqrt{n_{\mathrm{TF}}}$ (Eq. (5)) and approximating $\psi_0(\mathbf{r})^2$ by

$$\psi_0(\mathbf{r})^2 = \frac{n_{\mathrm{TF}}(\mathbf{r})}{2} \left( 1 + \frac{q}{2|g_s| n_{\mathrm{TF}}(\mathbf{r})} \right) \tag{16}$$

(see Eq. (3)), we find

$$\ddot{\mathbf{x}}_v \sim \frac{\xi_s^2}{\mu t_s^2} \left( 1 + \frac{q}{2|g_s| n_{\mathrm{TF}}} \right) \nabla U \Big|_{\mathbf{x}_v}. \tag{17}$$

The key insight provided by Eq. (17) is that a PCV will tend to accelerate in a direction of increasing potential, independent of the sign of the PCV charge, with a magnitude proportional to $|\nabla U|$. This is in stark contrast to the azimuthal motion of a scalar vortex. Equation (17) also predicts the acceleration will increase with $q$ due to an increase in $\psi_0^2$, as also evident from Eq. (13). Note that $|\nabla U|/\mu \sim 1/r_{F_\perp}$ and hence for large traps $r_{F_\perp} \gg \xi_s$ the PCV dynamics will occur over time scales $t \gg t_s$.

We also briefly remark that interactions with other PCVs could be included in Eq. (14) by including in $\nabla \theta$ the flow field of the additional PCVs. In a homogeneous system, this predicts $a \approx \psi_0(\mathbf{x}_v)^2/n_0$, with $a$ introduced empirically in [39,40] and related to the "spring constant" of the stretch energy.

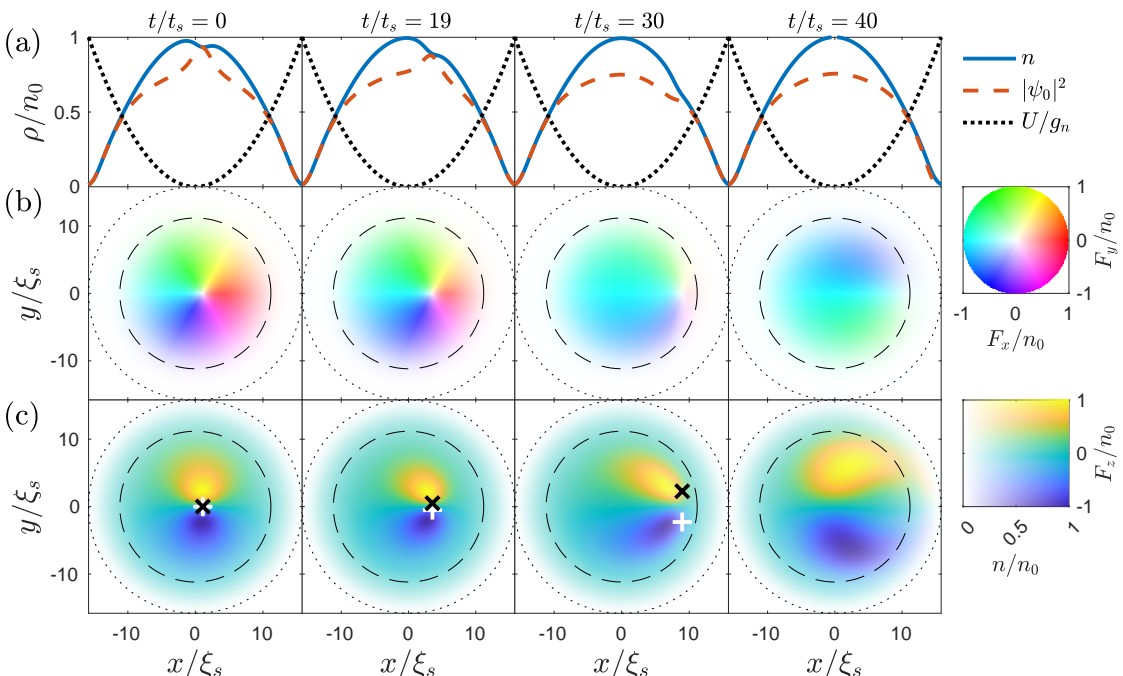

Figure 1: Dynamics of a PCV in a harmonic trap for $q = 0.5q_0$ and $\omega = 0.2q_0$. (a) Cross-sectional densities $\rho = n(x,0)$ (solid blue lines) and $\rho = |\psi_0(x,0)|^2$ (dashed red lines). The trap $\rho = U(x,0)/g_n$, Eq. (18), is shown for comparison (dotted black lines). (b) Transverse and (c) axial spin densities. The vortex core moves radially and also "stretches", which is associated with the formation of a dipole of $F_z$ magnetization. Dashed (dotted) circles in (b) and (c) are $r_{F_\perp}$ ($r_{\mathrm{TF}}$). White pluses (black crosses) in (c) mark centre of circulation of $\psi_1$ ($\psi_{-1}$) vortices.

## 4 PCV dynamics in a harmonic trap

We firstly explore PCV dynamics in a harmonic potential,

$$U(\mathbf{r}) = \frac{1}{2}M\omega^2 r^2. \tag{18}$$

A PCV at the centre of a harmonic potential is in unstable equilibrium. According to Eq. (17), an off-centre PCV will tend to move in the direction of increasing potential, i.e. to the condensate boundary. To test this, a positive, singly charged PCV is imprinted off-centre at $\mathbf{x}_v(0) = (\xi_s, 0)$ with wavefunction (see Eq. (3))

$$\Psi(\mathbf{r}) = \sqrt{\frac{n_{\mathrm{TF}}(\mathbf{r})}{2}} \begin{pmatrix} \sin\beta\, e^{-i\phi(\mathbf{r}-\mathbf{x}_v(0))} \\ \sqrt{2}\cos\beta \\ \sin\beta\, e^{i\phi(\mathbf{r}-\mathbf{x}_v(0))} \end{pmatrix}. \tag{19}$$

We then evolve the system using the spin-1 Gross-Pitaevskii equations Eq. (2) with strong damping ($dt \mapsto (1+i)dt$) for $-20t_s \leq t < 0$, allowing the density profile to stabilise and the PCV core to form. The component phase profiles are reset to the values in Eq. (19) throughout the damped evolution to ensure the PCV retains its position. Damping is then turned off and $\Psi$ is evolved conservatively using Eq. (2). Here and throughout the paper we simulate results on a $512 \times 512$ grid using fourth-order Runge-Kutta integration with kinetic energy operator evaluated to spectral accuracy. We use a system size of $100\xi_s \times 100\xi_s$ with $n_0 = 10^4\xi_s^{-2}$.

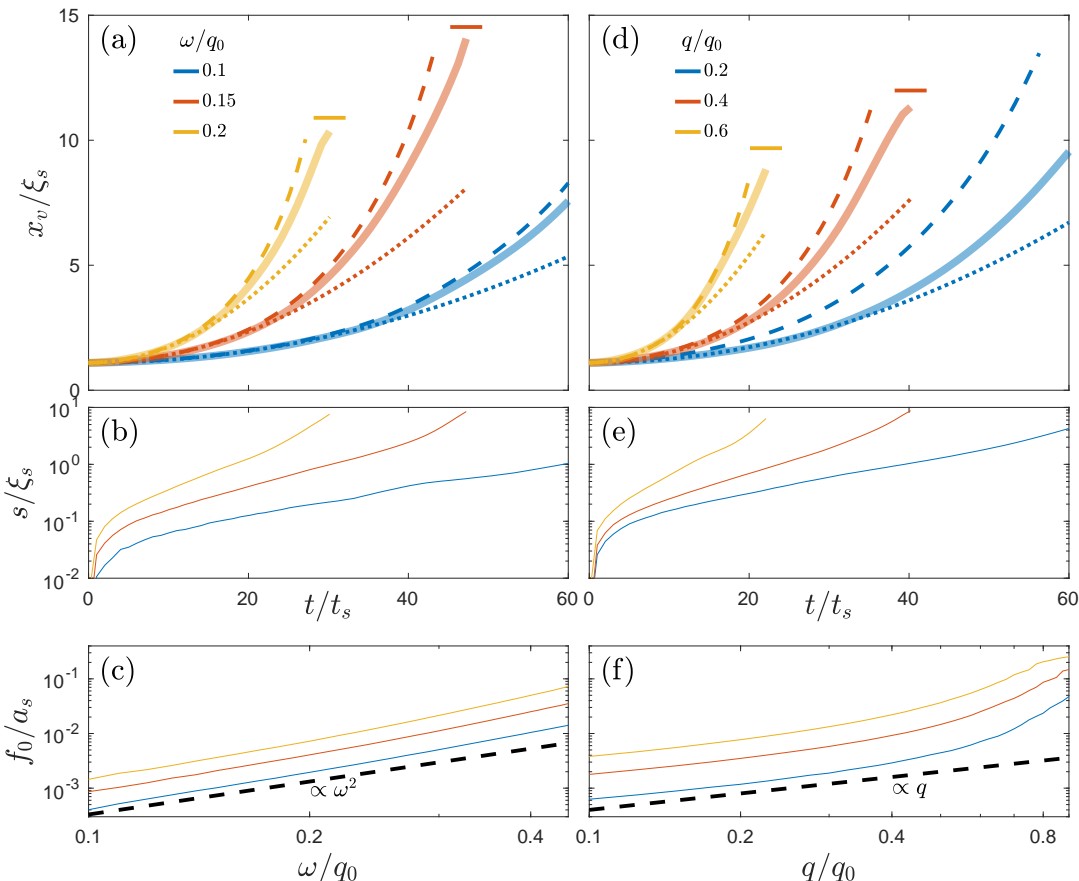

Figure 2:   (a) Evolution of PCV position for varying $\omega$ with $q = 0.5q_0$. Solid lines are numerical results, dashed lines are obtained from numerically integrating the right-hand side of Eq. (13) and dotted lines are $x_v(0) + \frac{1}{2}f_0 t^2$. Short horizontal lines mark $r_{F_\perp}$. (b) The corresponding PCV stretching. (c) The initial forcing $f_0$ obtained from the fits in (a) increases $\propto \omega^2$, consistent with Eq. (17). (d)-(f) As for (a)-(c) but for varying $q$ with $\omega = 0.2q_0$. The initial forcing increases $\propto q$, consistent with Eq. (17).

The dynamics of the PCV in the harmonic potential is shown in Fig. 1. The PCV accelerates radially outward, as predicted by Eq. (17). Associated with this, the vortex cores in the $\pm 1$ spin components separate orthogonal to the PCV motion, see Fig. 1(c). The PCV position and stretching are shown in Fig 2 for varying trap frequencies and quadratic Zeeman energies. The PCV acceleration is larger for larger trap frequency and larger quadratic Zeeman energy. Integrating the right-hand side of Eq. (13) using numerical data for **s** allows us to test the validity of Eq. (13). We find good agreement with the numerical results for $q \gtrsim 0.3q_0$, particularly for early times. We evaluate $|\psi_0(\mathbf{x}_v)|^2$ using Eq. (16), which gives a good estimate for the background component density. For later times, the stretch coordinate becomes large (see Fig 2(c),(d)) and hence we expect the approximation Eq. (10) to become less valid. The PCV radiates spin waves as it crosses $r_{F_\perp}$ and leaves the condensate, see Fig. 1. Note in some cases the PCV oscillates near $r_{F_\perp}$ due to interactions with the radiated spin waves before eventually leaving the condensate.

For early times $\dot{\mathbf{s}}$ is approximately constant, so that we may approximate $x_v$ by

$$x_v(t) \approx x_v(0) + \frac{1}{2}f_0 t^2 \quad \text{(early times)}, \tag{20}$$

see Fig 2(a),(b). Here $f_0$ is a constant forcing, obtained by fitting $x_v - x_v(0)$ to $(f_0/2)t^2$ across the first half of the PCV's motion. We find $f_0$ increases approximately quadratically with trap frequency and linearly with quadratic Zeeman energy, see Fig. 2(c),(f). This is consistent with the qualitative approximation Eq. (17). The increase in forcing with trap frequency follows directly from the scalar analysis, since the component vortices will circulate (stretch) more rapidly for tighter traps.

We find $f_0$ extrapolates to zero as $q \to 0$. Consistently, we find that Eq. (13) overestimates the PCV position for small $q/q_0$, see Fig. 2(d). The deviation may be caused by local spin rotations out of the transverse plane, which fundamentally alter the nature of the defect dynamics [70, 71]. An exploration of the stability of an off-centre PCV at $q = 0$ would reveal if the dynamics indeed does go to zero, or if it just slows down substantially.

# 5 PCV oscillations around a density minimum

The tendency for a PCV to move down density gradients means that PCV will be stable at a local potential maximum. Such a potential can therefore trap a PCV in principle. To explore the dynamics of this process, we examine the dynamics of a PCV in a potential

$$U(\mathbf{r}) = U_{\text{box}}(\mathbf{r}) + U_0 \exp\left(-\frac{r^2}{r_0^2}\right), \tag{21}$$

where $U_0 > 0$ and $r_0$ characterise the height and width respectively of the local potential maximum. The box trap $U_{\text{box}}(\mathbf{r}) = \mu \coth^{10}(1) \tanh^{10}(r/r_{\text{box}})$ characterised by size $r_{\text{box}} \gg r_0$ confines the condensate [72]. For $r_{\text{box}} \gg r_0$ we have $r_{\text{TF}} \approx r_{\text{box}}$. The potential Eq. (21) results in a local minimum in condensate density at the trap centre, see Fig. 3(a). A positive, singly charged PCV is imprinted at $\mathbf{x}_v(0) = (5\xi_s, 0)$ using Eq. (19). Further numerical details are the same as for the harmonic trap, see Sec. 4. The potential Eq. (21) increases radially inward for $r \lesssim r_{\text{box}}$, and hence the PCV initially moves radially inward, see Fig. 3.

Axial spin excitations are emitted as the PCV moves toward the trap centre. These excitations propagate out to and then reflect off the condensate boundary, see Fig. 3(c). The PCV stretching increases as the PCV velocity increases. Both the PCV position and stretching plateau near the trap centre. Rather than remaining here, however, the PCV position undergoes periodic kicks to a displacement on the order of the initial PCV position, see Fig. 4 (a), (c). This revival coincides with axial spin excitations localising close to the trap centre after reflecting off the condensate boundary. We interpret this as a re-excitation of the PCV motion due to absorption of axial spin excitations, which results in a rephasing of the transverse spin. Evidence for this is seen by increasing $r_{\text{box}}$, which increases the time for spin excitations to propagate to and from the condensate boundary, and hence the time $t_{\text{rev}}$ for the PCV displacement to revive, see Fig. 4(c). We find

$$\frac{t_{\text{rev}}}{t_s} \approx \frac{3r_{\text{box}}}{\xi_s} + 14, \tag{22}$$

see inset to Fig. 4(c), consistent with a constant spin-wave propagation speed of $v_s = 2r_{\text{box}}/t_{\text{rev}} \approx 0.7\xi_s/t_s$. This is close to the speed of sound of the gapless spin mode in a uniform system, which for $q = 0.5q_0$ is $v_s = 0.5\xi_s/t_s$ [73]. The small offset of 14 in Eq. (22) may account for the time for spin waves to interact with the PCV and reflect off the condensate boundary. The stretch coordinate of the PCV oscillates out of phase with the PCV displacement, acquiring a maximum value when the PCV is moving most rapidly, see Fig. 4(b).

The PCV continues to oscillate around the trap centre for long times $t \gg t_{\text{rev}}$, see Fig. 4(a). There is a weak decay in the oscillation amplitude with time, which we characterise by plotting

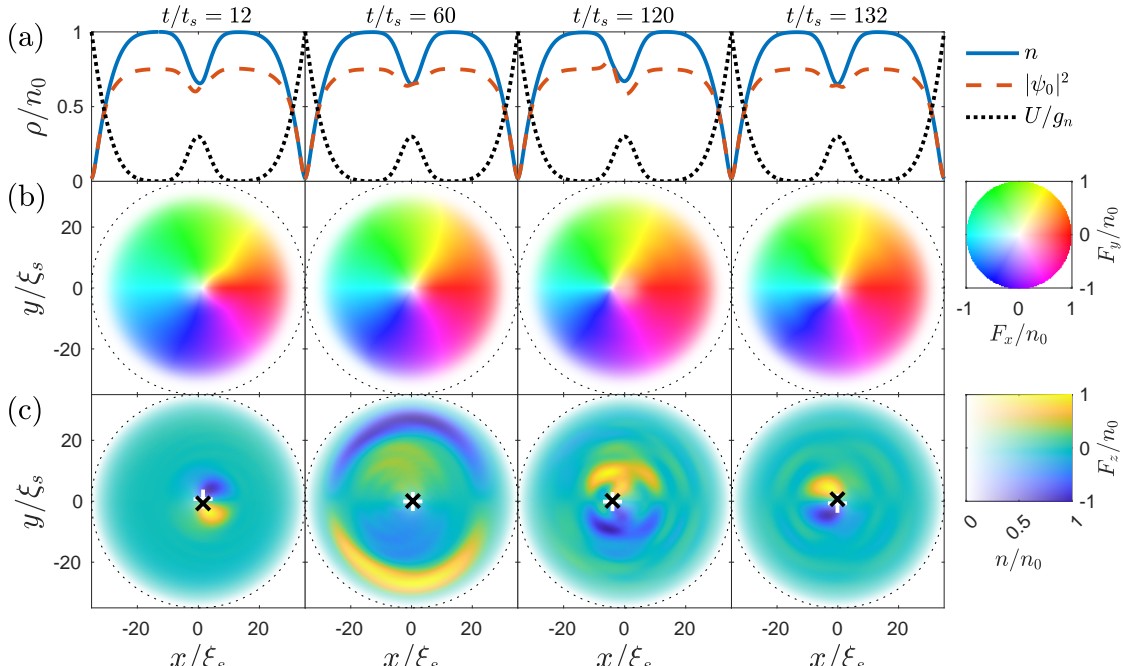

Figure 3: Dynamics of a PCV for the potential Eq. (21), with $q = 0.5q_0$, $r_0 = 5\xi_s$, $U_0 = 0.3\mu$ and $r_{\text{box}} = 35\xi_s$. (a) Cross-sectional densities $\rho = n(x, 0)$ (solid blue lines) and $\rho = |\psi_0(x, 0)|^2$ (dashed red lines). The potential $\rho = U(x, 0)/g_n$, Eq. (21), is shown for comparison (dotted black lines). (b) Transverse and (c) axial spin densities. The PCV moves down density gradients to the trap centre. Axial spin excitations are emitted, radiate to the condensate boundary, and then reflect back to displace the PCV again. Dotted circles in (b) and (c) are $r_{\text{box}}$. White pluses (black crosses) in (c) mark centre of circulation of $\psi_1$ ($\psi_{-1}$) vortices.

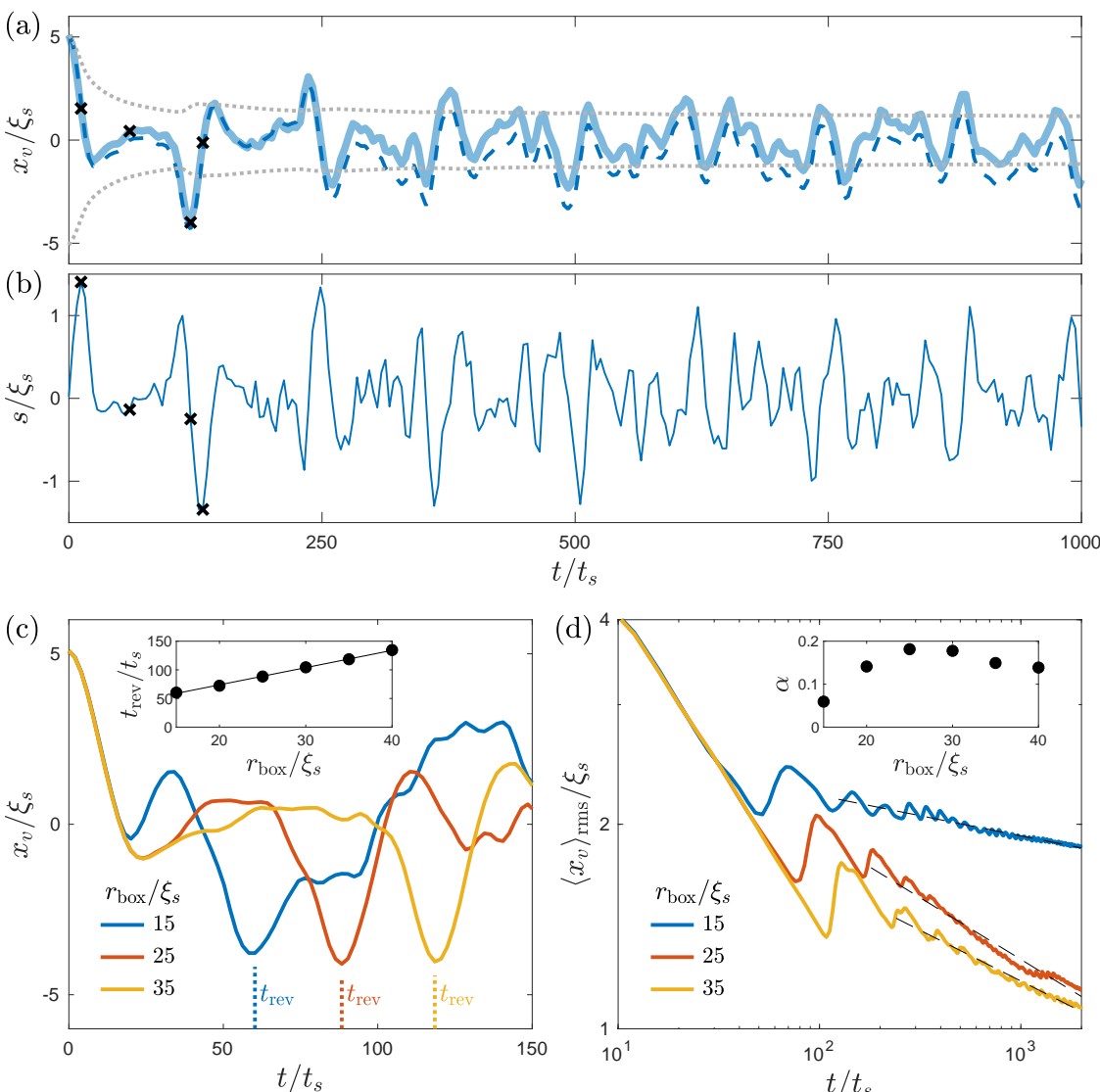

Figure 4: Evolution of (a) the PCV position and (b) PCV stretching, with parameters as in Fig. 3. The PCV position oscillates around the trap centre. Solid lines are the numerical results, dashed-blue line in (a) is obtained from numerically integrating the right-hand side of Eq. (13), crosses correspond to times in Fig. 3. The PCV stretching peaks when the PCV has a large velocity. (c) PCV dynamics showing oscillations with increasing box size. The magnitude of $x_v$ revives at a time $t_{\mathrm{rev}}$ (vertical dotted lines), which increases with box size. Inset: $t_{\mathrm{rev}}$ (circles) is fitted by $t_{\mathrm{rev}}/t_s \approx 3r_{\mathrm{box}}/\xi_s + 14$ (line), consistent with a constant axial spin wave propagation speed. (d) The rms amplitude of $x_v(t)$ (Eq. (23)) decays as $t^{-\alpha}$ at long times (black-dashed line) with exponent $\alpha$ dependent on box size (inset). Dotted-gray lines in (a) are $\pm\langle x_v\rangle_{\mathrm{rms}}/\xi_s$. All results are for $q = 0.5q_0$, $r_0 = 5\xi_s$ and $U_0 = 0.3\mu$.

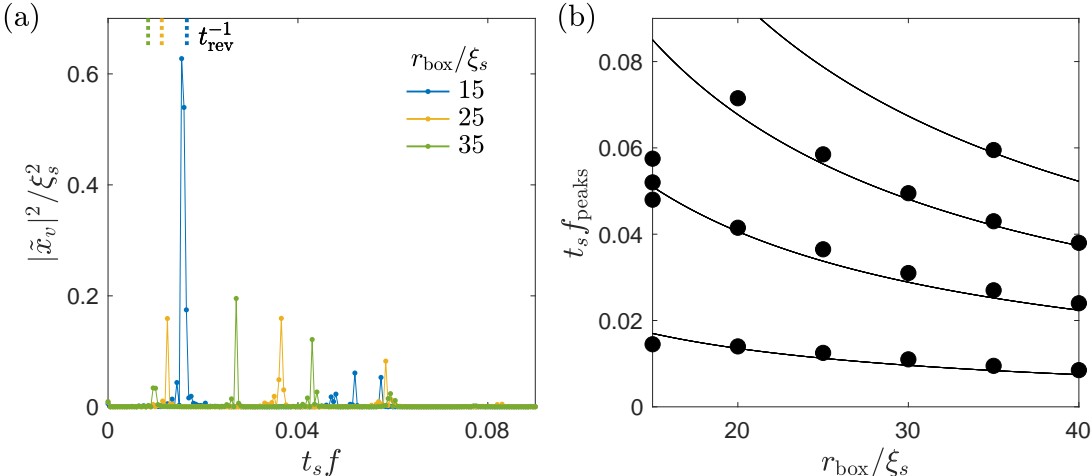

Figure 5: (a) The power spectra of $x_v$ (Eq. (24)) exhibit narrow, regularly spaced peaks corresponding to the oscillations in Fig. 4. The lowest of these corresponds to $t_{\text{rev}}^{-1}$ (vertical dotted lines). (b) The frequency of the peaks (circles) in $|\tilde{x}_v|^2$ decrease with increasing box size. Lines are $t_{\text{rev}}^{-1}$, $3t_{\text{rev}}^{-1}$, $5t_{\text{rev}}^{-1}$ and $7t_{\text{rev}}^{-1}$, with $t_{\text{rev}}$ given by Eq. (22). Only peaks with $|\tilde{x}_v|^2 \geq 0.02\xi_s^2$ and spaced $\geq 0.004t_s^{-1}$ from another peak are displayed. All results are for $q = 0.5q_0$, $r_0 = 5\xi_s$ and $U_0 = 0.3\mu$.

the rms amplitude

$$\langle x_v \rangle_{\text{rms}}(t) = \sqrt{\frac{1}{t} \int_0^t d\tau\, x_v(\tau)^2}, \tag{23}$$

see Fig. 4(d). We find $\langle x_v \rangle_{\text{rms}}$ can be fitted by a power law $At^{-\alpha}$ for long times $t > 2t_{\text{rev}}$, see black dashed line in Fig. 4(d). The decay exponent $\alpha$ is dependent on box size, with a larger box size tending to give rise to faster decay of oscillations, see inset to Fig. 4(d). A larger box size will have a larger phase space volume, hence enabling more rapid thermalization. The slow thermalization of axial spin waves is consistent with prior work on phase ordering [74].

Finally, we compute the power spectra of $x_v$, defined as

$$|\tilde{x}(f)|^2 = \left| 2T^{-1} \operatorname{Re} \int_0^T x_v(t) e^{-i2\pi f t}\, dt \right|^2, \tag{24}$$

with $T$ the evolution time. These are shown in Fig. 5(a). The power spectra exhibit narrow peaks at frequencies $f_{\text{peaks}}$ that decrease with increasing box size, see Fig. 5(b). The lowest frequency of these corresponds to $t_{\text{rev}}^{-1}$, consistent with Fig. 4(c). In addition there are peaks at $3t_{\text{rev}}^{-1}$ and $5t_{\text{rev}}^{-1}$ (and $7t_{\text{rev}}^{-1}$ for $r_{\text{box}} = 35\xi_s$). The generation of these higher frequency harmonics may be due to non-linear effects within the PCV, or due to non-linear interactions between the axial spin waves.

## 6 Energy considerations

The azimuthal motion of a vortex in a harmonically trapped scalar condensate ensures the kinetic energy contained in the flow field is conserved [67]. In contrast, the kinetic energy in

the flow field of a PCV is not conserved during its radial motion. In this section we explore the energy exchanges that occur during the PCV dynamics.

The contribution to the kinetic energy from the PCV flow field is

$$E_{\text{inc}} = \frac{\hbar^2}{2M} \sum_{m=-1,1} \int \mathrm{d}^2\mathbf{r} \, |\mathbf{v}_m^{\text{inc}}|^2, \tag{25}$$

where $\mathbf{v}_m^{\text{inc}}$ is the incompressible (divergence-free) component of the density-weighted velocity field $\mathbf{v}_m = |\psi_m|^{-1} \operatorname{Im}\left(\psi_m^* \nabla \psi_m\right)$ [75, 76]. Due to our initial condition, $\mathbf{v}_0 = 0$. Figure 6 shows that $E_{\text{inc}}$ decreases as the PCV centre — the region of highest fluid velocity — moves to lower density. The fluid flow also has a compressible component

$$E_{\text{com}} = \frac{\hbar^2}{2M} \sum_{m=-1,1} \int \mathrm{d}^2\mathbf{r} \, |\mathbf{v}_m^{\text{com}}|^2, \tag{26}$$

with $\mathbf{v}_m^{\text{com}}$ the compressible (curl-free) component of $\mathbf{v}_m$. We denote the remaining energy terms by

$$E_{\text{rem}} = H - E_{\text{inc}} - E_{\text{com}}. \tag{27}$$

In a harmonic trap, Eq. (18), $E_{\text{inc}}$ is monotonically converted predominantly into $E_{\text{rem}}$, see Fig. 6(a). The predominant contribution to $E_{\text{rem}}$ is the spin energy

$$E_s = q \int \mathrm{d}^2\mathbf{r} \left( |\psi_1|^2 + |\psi_{-1}|^2 + \frac{1}{2}|\mathbf{F}(\mathbf{r})|^2 \right), \tag{28}$$

see Fig. 6 (a), consistent with this energy arising due to PCV stretching. The oscillation in $E_s$ is likely due to collective modes in the system coupling both spin and density degrees of freedom.

In the potential Eq. (21), $E_{\text{inc}}$ oscillates with the oscillating PCV position, see Fig. 6(b). Here the times at which the PCV is close to the trap centre coincide with an average increase in $E_{\text{com}}$ rather than $E_{\text{rem}}$. This is consistent with the energy arising from radiated axial spin waves. Peaks of $E_{\text{inc}}$ occur when the PCV is at maximum displacement from the trap centre, and this coincides with troughs of $E_{\text{com}}$.

# 7   Conclusion

We have demonstrated that a PCV moves down density gradients in an inhomogeneous condensate. Our results are well-described by a simplified model, Eq. (13), adapted from a model of vortex dynamics in scalar condensates. Experiments using $^{87}$Rb have observed spin dynamics on the order of $10^2 \, t_s$ [41, 42], which is sufficient to observe the dynamics of a PCV in moderately confined harmonic traps. The recently observed spinor $^7$Li condensate has a much larger spin interaction strength $|g_s|$, enabling observation of spin dynamics on the order of $10^3 \, t_s$ [43, 44]. However, since $|g_s| \sim g_n$ in $^7$Li, density dynamics may be more significant. The PCV dynamics and axial spin excitations could be observed in experiments using *in situ* imaging [20–22, 24].

In asymmetric traps careful trap engineering [77] may allow steering of the PCV motion, resulting in the possibility of rich but controllable vortex trajectories. A non-zero net axial spin magnetization would provide a further control parameter, which we expect will allow for the vortex dynamics to be tuned continuously between PCV-like and scalar-like dynamics [40]. Finally, exploring the interplay between inhomogeneity and additional PCVs would be interesting, particularly in relation to nonlinear phenomena such as turbulence and the dynamics of phase transitions.

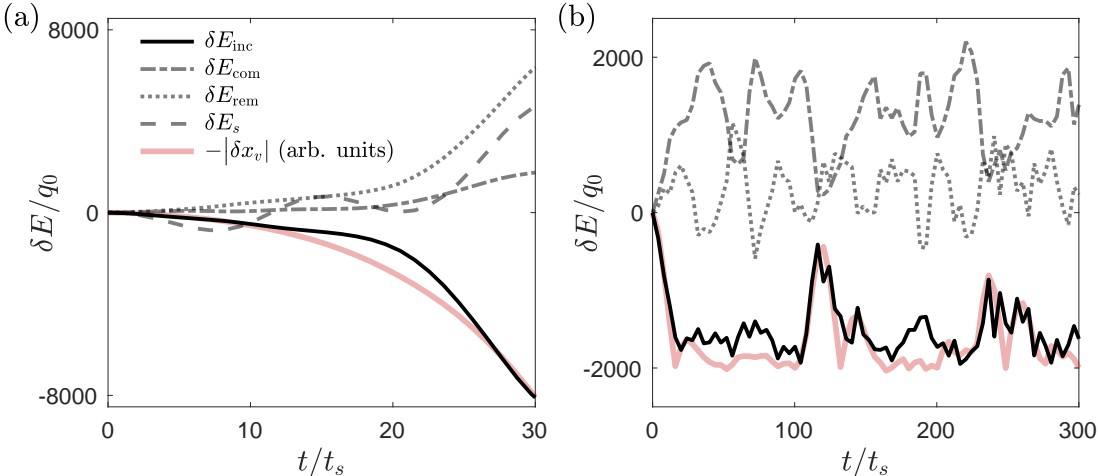

Figure 6: Energy exchanges during the PCV dynamics for both trapping potentials Eq. (18) and Eq. (21), with $\delta E_\mu(t) \equiv E_\mu(t) - E_\mu(0)$ and $\delta x_\nu \equiv |x_\nu(t)| - |x_\nu(0)|$. The incompressible energy $E_{\mathrm{inc}}$ decreases on average as the PCV moves to regions of lower density, with $\delta E_{\mathrm{inc}} \sim -|\delta x_\nu|$. (a) The harmonic trap, Eq. (18). Here $E_{\mathrm{inc}}$ is primarily converted into $E_{\mathrm{rem}}$ (predominantly $E_s$), with a small amount converted into $E_{\mathrm{com}}$ (parameters as in Fig. 1). (b) The potential with a local maximum at the centre, Eq. (21). Here $E_{\mathrm{inc}}$ is primarily converted into $E_{\mathrm{com}}$, consistent with the PCV radiating axial spin waves (parameters as in Fig. 3).

# Acknowledgements

We thank Andrew Groszek and Matthew Reeves for valuable discussions.

**Funding information** This research was supported by the Australian Research Council Centre of Excellence for Engineered Quantum Systems (EQUS, CE170100009).

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
