# Peer review of "Dynamics of Polar-Core Spin Vortices in Inhomogeneous Spin-1 Bose-Einstein Condensates"

_SciPost Physics_

## Round 1 · Referee Report · Anonymous (Referee 1) · 2024-5-20

Strengths

  1. Provides a path that can lead to further interesting models of the hydrodynamics of spinor Bose-Einstein condensates.
  2. The results can in principle inspire immediate experiments using existing setups.
  3. The manuscript is focussed and well written.

Weaknesses

  1. The referencing is possibly a little narrowly focussed and while it captures the immediately important works may miss some relevant context from other recent experiments.

  2. There is a small number of typographical errors in the manuscript, including inline equations overflowing into the margins and one or two instances of strange spacing in inline equations. This is not any scientific criticism, but included here since the journal specifically asks about paper formatting.

Report

In their manuscript, the authors give a detailed description of what they term a polar-core spin vortex (PCV) in a spin-1 Bose-Einstein condensate (BEC) in two experimentally relevant contexts: a harmonically trapped condensate and a condensate in a box trap with a pinning potential. They find that in the former case, the vortex moves down the density gradient without exhibiting the familiar precession of a vortex in a scalar BEC. In the latter case, the vortex is found to oscillate around the pinning site, exibiting stretching of the vortex-core region. The study is interesting and well presented and should be of interest for both theorists and experimenters. One may quibble whether the study really represents "a groundbreaking theoretical/computational discovery", but it certainly presents a pathway for analysing spinor-vortex dynamics that could be built upon, e.g., to understand spinor vortex interactions, leading to hydrodynamic models. For this reason I support publication, provided the authors consider a number of relatively minor questions.

Requested changes

  1. The authors work in what they call the easy-plane phase of a condensate with ferromagnetic interactions. This corresponds to the broken-axisymmetry phase of Ref. 25 at zero longitudinal magnetisation such that the condensate spin has no z-component. The authors correctly state that the condensate spin becomes density dependent, such that the condensate reaches the polar phase beyond some radius in the harmonic trap. What is the spin magnitude at the centre of the trap/at the initial position of the vortex.

  2. In the easy-plane phase, with the condensate spin confined to the x,y-plane, one can easily see that the order-parameter space should correspond to U(1) x SO(2). In this case, the vortex with no U(1) charge and a 2pi winding of the in-plane spin vector, leading to a first homotopic group Z x Z and the pure spin vortices considered here. However, it is clear from the presented simulations that the condensate spin does not remain fully in the x,y-plane, but rather develops a non-negligible z-component. How does this modify the understanding of the nature of the vortices? Do they pick up a mass circulation?

  3. For the same reason, once the vortex core stretches and the regions of non-zero F_z appear, beta in Eq. (19) aquires a spatial dependence. Can the the vortex spinor be brought back on this form through a change of spinor basis, similar to the corresponding ferromagnetic vortex in Lovegrove et al., PRA 86, 013613 (2012)?

  4. The authors consider a pinning potential together with the box trap, but not the harmonic trap. While box traps are becoming increasingly interesting for experiments, the harmonic trap remains the common choice. Should one expect the results observed for the pinning in the box trap to hold also if one were to introduce a pinning potential in a harmonic trap?

  5. The authors begin their numerical simulation by including "a strong damping dt -> (1+i)dt", i.e., equal real and imaginary parts for the time step. Is there a reason this step is not done as relaxation in imaginary time (with no real part)? The authors then proceed to integrate the GPEs forward in time with no dissipation. How do results change if a small dissipation is included (i.e., complex time dt -> (1 + eta i)dt for eta ~10^-2 or 10^-3, as a rough model of the small dissipation present in experiments?

  6. In their introduction, the authors cite Refs. 4-11 for vortex pinning. However, not all of these seem to be examples of topological defects (but rather, e.g., classical vortices) as implied by the context of the rest of the beginning of the sentence. Can the authors clarify?

  7. The immediately relevant spinor-BEC experiments using in situ detection in the group of Y.-i. Shin are appropriately cited in the introduction. However, there have been several other experimental realisations of different types of spinor vortices in the last several years, including in condensates with ferromagnetic interactions (such as Rb-87), for which the results of the present manuscript may be equally relevant, and which ought to provide relevant context.

Finally a very minor detail:

  1. The authors model the box trap using the expression given below Eq. (21). How well does this tanh^{10} form model existing box traps such as that of Ref. 72? The expression also includes a constant factor coth^{10}(1). There is presumably a reason for writing this constant on this form?

Recommendation

Ask for minor revision

  • validity: high
  • significance: good
  • originality: top
  • clarity: high
  • formatting: good
  • grammar: perfect

Author:  Lewis Williamson  on 2024-10-23  [id 4896]

(in reply to Report 1 on 2024-05-20)

We thank the referee for their careful reading of our manuscript and endorsement of our work. Below we respond to each of the referee's numbered points. Equation numbering refers to the updated manuscript.

  1. The transverse spin density tends to decrease moving from the trap centre to the trap boundary, with a sharp dip at the PCV core. We have included plots of $F_\perp$ along the line of vortex motion in Fig. 1(a) and Fig. 3(a). Note this spin density is also apparent in Fig 1(b) and Fig 2(b). Plotting $|F|$ is the same, since $F_z$ is zero along this cross-section.

  2. The topology of the system is preserved under the evolution. The axial spin that develops does not affect mass flow, as the net contribution from the $m=+1$ and $m=-1$ components cancel. This would change if there was an imbalance of $m=+1$ and $-1$ components, i.e. a net total axial magnetization. Alternatively, it is conceivable that a mass flow could develop if $q\rightarrow 0$ with appropriate symmetry breaking perturbations, as here the order parameter space is SO(3) and mass and spin vortices are topologically equivalent. We have clarified that the total mass current remains zero with new text below Eq.~(12).

  3. The splitting of the component circulations that make up a PCV occurs within the vortex core. Here the state deviates from the order parameter space in a complicated, spatially dependent way. We do not think it would be worthwhile to consider basis transformations within this region, as we see no clear way to restrict the possible transformations to a subset of SU(3) (unlike in the Lovegrove et al. paper above, where the transformations belong to SO(3)). Exploring the core structure of a stretched PCV would no doubt be interesting, but is beyond the scope of this work. We have clarified that Eq.~(4) does not describe the stretched PCV (new text below Eq.~(10)) and have highlighted that exploring the non-trivial core structure that develops during the dynamics would be an interesting area for further research in the conclusion.

  4. Qualitatively, the results are similar. In general, softening the trap walls (for example using a harmonic trap) results in axial spin wave dynamics and resulting vortex oscillations that are more complicated. Furthermore, collective excitations (breathing and higher order modes) of the condensate may be excited. This complicates the quantitative analysis. We have included text explaining the effect of trap softness at the end of Sec.~(5).

  5. Our method to obtain the initial state was purely convenience as it required us to use existing code. We have now modified our method to use imaginary time evolution to avoid confusion. Our results are robust to weak damping $\gamma\lesssim 10^{-3}$. The effects of larger damping are interesting and tends to slow the vortex motion for $10^{-2}\lesssim \gamma\lesssim 1$ and accelerate the vortex motion for $\gamma>1$. We have modified our results so that the initial state is obtained using purely imaginary time evolution, to avoid confusion. We have added new text to the conclusion discussing the effects of damping on the dynamics.

  6. We agree that some of Refs [4-11] may not be appropriate and thank the referee for identifying this. We have removed Refs [10] and [11] from the list [4-11] and also changed turbulence to quantum turbulence in the first sentence.

  7. We have included additional references to experimental works realising spin defects and textures, see updated references in [18-23].

  8. The constant factor $\coth^{10}(1)$ is included as it ensures $U(r_\mathrm{box})=\mu_\mathrm{TF}$ and hence $r_\mathrm{TF} \approx r_\mathrm{box}$. The form $\tanh^{10}$ has a slightly softer wall than the steep power law used to model experiments discussed in Ref.~[72], however using a steeper wall gives comparable results. A $\tanh(x)$ function is often used to model box traps, as it does not require imposing a sharp maximum size to the trap; a power law trap requires imposing a maximum size to avoid numerical issues with the walls growing rapidly in size. We have added text below Eq. (22) clarifying why we include the constant factor $\coth^{10}(1)$.

---

## Round 1 · Referee Report · Anonymous (Referee 2) · 2024-5-21

Strengths

  1. Nice study of potential relevance to current/future experiments with spinor atomic condensates.

  2. Combined numerical study with supporting analytical model providing further qualitative understanding.

  3. Clearly written manuscript with nice figures and extended referencing.

Weaknesses

  1. Some findings could be better analysed/quantified.

  2. I am generally not in favour of short-hand notations ("PCV" used here for polar core vortex).

Report

This is an interesing paper discussing the dynamics of polar-core spin vortices in spinor atomic Bose-Einstein condensates. Main results are that such a vortex structure will accelerate radially towards the edge of a harmonically-trapped condensate, whereas emission and reabsorption of axial spin waves leads to oscillations about the trap centre in a trap with a local density minimum. Numerical simulations based on mean-field theory (Gross-PItaevskii) are consistent with a simpler point-vortex-based model, which further explains the underlying physical process.

While such work is not really ground-breaking, it can nonetheless be seen as opening a new experimentally-relevant direction in vortex dynamics in spinor condensates: as such, I am happy to support its publication, subject to a few further comments/clarifications.

Requested changes

  1. The authors use a strongly dissipative Gross-Pitaevskii model for the initial state, followed by Gross-Pitaevskii dynamical propagation: it was not clear to me why they chose to include (1-i) as opposed to simply -i for the initial relaxation. Moreover, given that the code has such direct capability, I wonder whether the authors tried adding a small damping of the form (1-i gamma) [e.g. with a small gamma value in the typical range 10^(-3)-10^(-2) or so] to investigate to what extent their findings are significantly affected, and what the qualitative nature of the resulting changes would be to the dynamical evolution of such a vortex.

  2. The authors choose to work in the parameter regime g_n/|g_s| = 10 to focus on the spin degrees of freedom. While fully acceptable, could they comment more on how a different choice might affect their findings (at the very least qualitatively)?

  3. Is there any reason why the t=0 dynamics is not included in Fig. 3 (as it is in Fig. 1)? I feel it might be helpful. Also, perhaps a zoom-in of the region near x=0 might be useful to include, to better highlight the +5 xi_s initial condition (unless I am missing something here).

  4. Moreover, might it be useful to more clearly highlight what is plotted, in terms of the +1 and -1 densities?

  5. What is the specific role of the boundaries, besides acting so as to reflect the emitted waves, and are the findings at all sensitive to the way they are modelled? In particular, would a different choice of underlying trap (e.g. harmonic trap with a Gaussian barrier in the middle) produce qualitatively very similar results, or would the process be affected by distinct propagation of emitted waves in such an underlying trap?

  6. Some more clarifications around Eq. (22) would be welcome: are these values somehow anticipated, or purely fitting extracted? If the latter, is this a unique parameter choice, and can those numbers be more physically interpreted?

  7. I was slightly confused about the changing value of the decay exponent alpha after Eq. (23). While I find such analysis, and discussion of the box dependence useful, could something more be extracted from this? Are these and earlier results insensitive to the initial condition choice (e.g. polar-core vortex location?)

  8. In single-component condensates, vortex motion has been carefully characterised in 2d. Would the authors expect to ultimately find similar controlled monitoring in spinor systems, or are there genuine challenges prohibiting that? If so, perhaps it might be useful to give some plausible experimental numbers for such observation, as an indication. The one aspect I am unsure of is the survival timescale of such defects, so it would be interesting to see (presumably in a future publication) the effect that some small fluctuations in the initial condition might actually have on the ensuing motion.

Recommendation

Publish (meets expectations and criteria for this Journal)

  • validity: high
  • significance: good
  • originality: high
  • clarity: high
  • formatting: excellent
  • grammar: excellent

Author:  Lewis Williamson  on 2024-10-23  [id 4897]

(in reply to Report 2 on 2024-05-21)

We thank the referee for their careful reading of our manuscript and endorsement of our work. Below we respond to each of the referee's numbered points. Equation numbering refers to the updated manuscript.

  1. Our method to obtain the initial state was purely convenience as it required us to use existing code. We have now modified our method to use imaginary time evolution to avoid confusion. Our results are robust to weak damping $\gamma\lesssim 10^{-3}$. The effects of larger damping are interesting and tends to slow the vortex motion for $10^{-2}\lesssim \gamma\lesssim 1$ and accelerate the vortex motion for $\gamma>1$. We have modified our results so that the initial state is obtained using purely imaginary time evolution, to avoid confusion. We have added new text to the conclusion discussing the effects of damping on the dynamics.

  2. Our conclusions for the harmonic trap remain unchanged for $g_n/|g_s|=5$ and hence we expect the results to be representative of cases $|g_s|\ll g_n$ such as in $^{87}$Rb. For $g_n/|g_s|\approx 2$, as occurs for $^7$Li, the background condensate profile and the quasiparticle excitations will be affected. We have modified the text below Eq.~(3) to clarify that we expect our results to be representative of cases $|g_s|\ll g_n$. We have added new text to the conclusion discussing the possible changes that would occur for $|g_s|\sim g_n$, as occurs in $^7$Li.

  3. We prefer to omit a zoom in due to space limitations. We believe that quantitative details of the vortex position are already sufficiently clear from Fig 4. We have now included a $t=0$ frame in Fig 3.

  4. We have added explicit expressions for the spin densities in terms of spin components, see new Eq (2).

  5. Qualitatively, the results are similar. In general, softening the trap walls (for example using a harmonic trap) results in axial spin wave dynamics and resulting vortex oscillations that are more complicated. Furthermore, collective excitations (breathing and higher order modes) of the condensate may be excited. This complicates the quantitative analysis and hence we prefer not to explore this here. We have included text explaining the effect of trap softness at the end of Sec.~(5).

  6. The parameters in this equation are fitted and the fit is unique. The physical origin of these is discussed below Eq. (23) but we appreciate this may not have been clear. The fitted velocity is close to the spin wave propagation speed and we speculate that the offset time may arise from the time spin waves take to interact with the PCV and reflect off the condensate boundary. We have clarified that the parameters are fits by relabelling them $v_\mathrm{fit}$ and $t_\mathrm{offset}$ and have clarified their physical interpretation below Eq. (23).

  7. Looking into this, we find that this trend is indeed sensitive to the initial vortex position. We find that $\alpha$ tends to decrease as the initial vortex position is moved closer to the trap centre and so we have removed a mention of trend on box size. We have removed a mention of the trend of $\alpha$ on box size and included text below Eq.~(24) clarifying the dependence on initial vortex position.

  8. Current experiments using $^{87}$Rb have lifetimes long enough for observable vortex dynamics to occur in a trap. This, combined with \emph{in situ} imaging techniques, should enable experimental observation of the dynamics. This is discussed in the conclusion. The suggestion of exploring effects of noise is an interesting are for future work. We have now mentioned the possibility of exploring finite-temperature dynamics in the conclusion, as part of new text discussing effects of damping.

---

## Round 1 · Referee Report · Anonymous (Referee 3) · 2024-5-22

Strengths

The authors show the qualitative differences in the dynamics of different topological vortices.

Report

In this manuscript, the authors have investigated the dynamics of polar-core vortices in a spin-1 ferromagnetic Bose-Einstein condensate (BEC) with non-uniform density. The polar-core vortex is the vortex with spin singularity. Under an easy-plane anisotropy due to the quadratic Zeeman effect, the $m=1$ and $-1$ components have opposite phase winding at the same position with the $m=0$ component filling the core. The authors consider the motion of such a polar-core vortex under a non-uniform density distribution in a confining potential. In the case of a scalar BEC, it is known that the inhomogeneous density profile induces an effective force to the vortices, where vortices with opposite circulation rotate in the opposite direction around the trap center. Thus, the vortices in the $m=1$ and $-1$ components move in the opposite direction. In addition, differently from the case of scalar BEC, the vortices feel an additional force due to the spin interaction. The authors have analytically derived the equation of motion of the relative and mean positions of the vortices in $m=\pm1$ components. The result shows that the polar-core vortex moves to the lower-density region, which the authors confirm by numerical calculation. In the case of a simple harmonic trap, the vortex goes out of the condensate. Then, the authors introduce a density dip at the trap center and show that the vortex is pinned and oscillates around the density dip.

This work extends the analysis of vortex dynamics in a scalar BEC to a spinor one. The authors clearly show the difference in the behavior of the phase vortex in a scalar BEC and the polar-core vortex in a spinor BEC. The fact that such a difference in the topology of vortices leads to different dynamics sounds quite interesting, and it is of interest to a wide range of researchers concerning topology. Although several points are unclear (see below), I’d like to recommend the publication of this manuscript in SciPost Physics.

Requested changes

1. The authors consider the motion of phase singularities in the $m=\pm1$ components and show that the relative position of the vortices becomes larger as the mean position goes down the density gradient. This reads as if the polar-core vortex is unstable against the splitting. However, the polar-core vortex should be stable even in the setup the authors considered. The polar-core vortex should be defined as a singularity of magnetization, so the position of the polar-core vortex can be traced at least in the numerical simulation. The authors should clarify this point in the manuscript. I feel showing the profiles of $|{\bf F}|/n$ is informative.
2. The separation of the two vortices in the $m=\pm1$ components is the appearance of the longitudinal magnetization around the polar core. What is the origin of the longitudinal magnetization? How does the magnetization flow under a density gradient and the spin current around the vortex?
3. I could not understand how Eq. (9) is derived.
4. $\kappa$ seems to be restricted to $\pm1$ after Sec. 3. This should be clarified.

Recommendation

Ask for minor revision

  • validity: good
  • significance: good
  • originality: good
  • clarity: ok
  • formatting: good
  • grammar: excellent

Author:  Lewis Williamson  on 2024-10-23  [id 4898]

(in reply to Report 3 on 2024-05-22)

We thank the referee for their careful reading of our manuscript and endorsement of our work. Below we respond to each of the referee's numbered points. Equation numbering refers to the updated manuscript.

  1. The stability of a PCV in motion is an interesting question and was tentatively explored in ref [40]. A moving PCV is stretched and for large stretching (large PCV velocity) this can destabilise the vortex while still preserving the total topology. We do not explore this regime in the present paper, however it is an interesting area for future research. The position $\mathbf{x}_v$ of the PCV coincides with the centre of circulation of $F_\perp$, which can be traced in the numerical simulation. We have included plots of $F_\perp$ along the line of vortex motion in Fig. 1(a) and Fig. 3(a). Note this spin density is also apparent in Fig 1(b) and Fig 2(b). Plotting $|F|$ is the same, since $F_z$ is zero along this cross-section. We have add new text to the conclusion discussing the potential instability that arises for a fast-moving vortex.

  2. Longitudinal magnetization arises when there is an imbalance in plus and minus spin components, see new Eq (2) in the paper. As the two vortices in $m=\pm 1$ separate, it becomes energetically favourable for the core of the $m=+1$ vortex to be partially filled by the $m=-1$ component and vice versa, since this lowers the spin interaction energy. Hence the $F_z$ magnetization develops a dipole structure. Exploring how the axial spin tends to flow in the presence of density gradients or spin currents is an interesting question but beyond the scope of our work. We have clarified the origin of $F_z$ magnetization, see new text in the second paragraph of Sec 4.

  3. This equation follows from symmetry under the transformation $(F_x,F_y,F_z)\rightarrow (F_x,-F_y,-F_z)$, which changes the vortex sign and interchanges the $m=\pm 1$ components. We have added text describing this symmetry in a footnote at the bottom of page 4.

  4. This is correct. We have clarified this in the text above Eq (4) and at the start of Sec 3.

---

## Round 2 · Referee Report · Anonymous (Referee 3) · 2024-11-19

Report

The authors have sincerely responded to the questions I raised before.
However, I still do not understand the answer to question 3: the derivation of Eq. (10).
The authors say that it follows from symmetry under the transformation (Fx, Fy, Fz) -> (Fx, -Fy, -Fz). Why is the order parameter symmetric under this transformation? When we choose m=1, for example, Eq. (10) reads |ψ1(x1,t)|2=|ψ1(x1,t)|2. This may hold in some symmetric configurations as the authors numerically simulated. However, I think it does not generally hold, in particular, in an inhomogeneous condensate. I guess Eq. (10) is also related to the approximation θ1|x1θ1|x1, which comes from choosing ψ0(r) to be real.
I understand the result in this paper is correct: I expect |ψ1(x1,t)|2=|ψ1(x1,t)|2 holds when s is always perpendicular to the density gradient, and indeed the authors show ˙slnA. I would like to ask the authors to derive Eq. (10) explicitly or to clarify the assumptions if the authors impose some assumptions.

Recommendation

Publish (easily meets expectations and criteria for this Journal; among top 50%)

---

## Round 2 · Referee Report · Anonymous (Referee 1) · 2024-11-30

Strengths

1. Nice study of potential relevance to current/future experiments with spinor atomic condensates.

2. Combined numerical study with supporting analytical model providing further qualitative understanding.

3. Clearly written manuscript with nice figures and extended referencing.

Weaknesses

Previously identified weaknesses addressed in the revised manuscript; hence N/A.

Report

I would like to thank the authors for their thoughtful replies to the questions and comments posed in my initial report. Additional analysis has been supplied where appropriate and feasible, and the authors have provided reasonable explanations and justifications where outside the scope or unfeasible. I agree with the authors that the cores structure of the deformed vortex would be interesting to analyse but that this may be deferred for future work. I am well satisfied that my comments have otherwise been suitably addressed and I think the revisions to figures and text make for significant improvements in clarity. In conclusion, I recommend publication of the manuscript in its present form.

Requested changes

None.

Recommendation

Publish (easily meets expectations and criteria for this Journal; among top 50%)

---

## Round 2 · Referee Report · Anonymous (Referee 2) · 2024-12-2

Strengths

1. Nice study of potential relevance to current/future experiments with spinor atomic condensates.

2. Combined numerical study with supporting analytical model providing further qualitative understanding.

3. Clearly written manuscript with nice figures and extended referencing.

Weaknesses

None

Report

I thank the authors for their detailed responses.
The amended manuscript can now be accepted.

Recommendation

Publish (easily meets expectations and criteria for this Journal; among top 50%)

---

## Round 2 · Author Response

We sincerely thank all three referees for their careful reading of our manuscript and constructive comments. We have addressed all queries raised by the referees. With these changes, we believe our manuscript is now ready for publication.

---

## Round 2 · List of Changes

We have included plots of F along the line of vortex motion in Fig. 1(a) and Fig. 3(a).

We have clarified that the total mass current remains zero with new text below Eq.~(12).

We have clarified that Eq.~(4) does not describe the stretched PCV (new text below Eq.~(10)) and have highlighted that exploring the non-trivial core structure that develops during the dynamics would be an interesting area for further research in the conclusion.

We have included text explaining the effect of trap softness at the end of Sec.~(5).

We have modified our results so that the initial state is obtained using purely imaginary time evolution. We have added new text to the conclusion discussing the effects of damping on the dynamics.

We have removed Refs [10] and [11] from the list [4-11] and also changed turbulence to quantum turbulence in the first sentence.

We have included additional references to experimental works realising spin defects and textures, see updated references in [18-23].

We have added text below Eq. (22) clarifying why we include the constant factor coth10(1).

We have modified the text below Eq.~(3) to clarify that we expect our results to be representative of cases |gs|gn. We have added new text to the conclusion discussing the possible changes that would occur for |gs|gn, as occurs in 7Li.

We have included a t=0 frame in Fig 3.

We have added explicit expressions for the spin densities in terms of spin components, see new Eq (2).

We have clarified that the parameters are fits by relabelling them vfit and toffset and have clarified their physical interpretation below Eq. (23).

We have removed a mention of the trend of α on box size and included text below Eq.~(24) clarifying the dependence on initial vortex position.

We have mentioned the possibility of exploring finite-temperature dynamics in the conclusion, as part of new text discussing effects of damping.

We have add new text to the conclusion discussing the potential instability that arises for a fast-moving vortex.

We have clarified the origin of Fz magnetization, see new text in the second paragraph of Sec 4.

We have added text describing this symmetry in a footnote at the bottom of page 4.

We have clarified that κ is restricted to ±1 in the text above Eq (4) and at the start of Sec 3.

---

## Round 3 · Author Response

List of changes
We have improved Sec. 3 to more clearly justify the equations of motion Eq. (18) and (19).
We have improved the qualitative estimate Eq. (23) by more accurately accounting for the background field densities.
We have modified the discussion below Eq. (26) to be consistent with our changes to Sec. 3.
We have made a minor aesthetic change to Fig. 2(b),(e).

---

## Round 3 · List of Changes

We have improved Sec. 3 to more clearly justify the equations of motion Eq. (18) and (19).
We have improved the qualitative estimate Eq. (23) by more accurately accounting for the background field densities.
We have modified the discussion below Eq. (26) to be consistent with our changes to Sec. 3.
We have made a minor aesthetic change to Fig. 2(b),(e).

---

## Editorial Decision

editorial_decision: